Cranial and mandibular shape variation in the genus Carollia (Mammalia: Chiroptera) from Colombia: biogeographic patterns and morphological modularity

López-Aguirre Camilo 1 2 cernesto.laguirre@gmail.com
Pérez-Torres Jairo 3
Wilson Laura A. B. 1
1 School of Biological, Earth, and Environmental Sciences, University of New South Wales , Sydney , Australia
2 Unidad de Ecología y Sistemática (UNESIS), Departamento de Biología, Pontificia Universidad Javeriana , Bogotá , Colombia
3 Laboratorio de Ecología Funcional, Unidad de Ecología y Sistemática (UNESIS), Departamento de Biología, Pontificia Universidad Javeriana , Bogotá , Colombia
Abdala Virginia
Electronic publication date: 2015 Aug 13
Publication date: 2015
Volume: 3
Electronic Location ID: e1197
Received 2015 May 19; Accepted 2015 Jul 28
Copyright: © 2015 López-Aguirre et al.
Copyright year: 2015
Copyright holder: López-Aguirre et al.
License: This is an open access article distributed under the terms of the Creative Commons Attribution License, which permits unrestricted use, distribution, reproduction and adaptation in any medium and for any purpose provided that it is properly attributed. For attribution, the original author(s), title, publication source (PeerJ) and either DOI or URL of the article must be cited.
License URL: https://creativecommons.org/licenses/by/4.0/

Keywords: Phyllostomidae, Neotropics, Ecomorphology, Limiting similarity, Functional morphology

Funding: The authors received no funding for this work.

==============================
Neotropical bats of the genus Carollia are widely studied due to their abundance, distribution and relevance for ecosystems. However, the ecomorphological boundaries of these species are poorly differentiated, and consequently correspondence between their geographic distribution, ecological plasticity and morphological variation remains unclear. In this study, patterns of cranial and mandibular morphological variation were assessed for Carollia brevicauda, C. castanea and C. perspicillata from Colombia. Using geometric morphometrics, morphological variation was examined with respect to: differences in intraspecific variation, morphological modularity and integration, and biogeographic patterns. Patterns of intraspecific variation were different for each species in both cranial and mandibular morphology, with functional differences apparent according to diet. Cranial modularity varied between species whereas mandibular modularity did not. High cranial and mandibular correlation reflects Cranium-Mandible integration as a functional unit. Similarity between the biogeographic patterns in C. brevicauda and C. perspicillata indicates that the Andes do not act as a barrier but rather as an independent region, isolating the morphology of Andean populations of larger-bodied species. The biogeographic pattern for C. castanea was not associated with the physiography of the Andes, suggesting that large body size does not benefit C. brevicauda and C. perspicillata in maintaining homogeneous morphologies among populations.

Introduction

Morphological innovation plays a central role in the speciation and diversification of mammals (Dumont et al., 2012). This feature has allowed mammals to develop vast ecomorphological diversity, making them one of the most efficient vertebrate groups in terms of both colonising and specialising to new environments (Venditti, Meade & Pagel, 2011). Among mammals, the family Phyllostomidae (Chiroptera) has undergone considerable adaptive radiation, occupying a wide variety of ecological niches associated to diet, comprising frugivorous, insectivorous, nectarivorous, carnivores, and hematophagous guilds (Dumont, 1997). Among these guilds, frugivory is the most related to morphological innovation and ecological diversification in phyllostomid bats (Freeman, 2000). Of all phyllostomid bats, frugivorous species display one of the highest degrees of morphological plasticity (Dumont, 1997; Rojas et al., 2012).

Bats grouped into the genus Carollia are important for ecosystems as seed dispersers and pollinators, owing to their diet, abundance and distribution (Muscarella & Fleming, 2007). The genus Carollia comprises eight species, of which two are restricted to Central America (Wright et al., 1999; Zurc & Velazco, 2010): C. sowelli (Baker, Solari & Hoffman, 2002) and C. subrufa (Hahn, 1905); three to South America: C. manu (Pacheco, Solari & Velazco, 2004), C. monohernandezi (Muñoz, Cuartas & Gonzáles, 2004) and C. benkeithi (Solari & Baker, 2006); and three distributed in both: C. brevicauda (Schinz, 1821), C. castanea (Allen, 1890) and C. perspicillata (Linnaeus, 1758). In Colombia—which has the highest phyllostomid species richness in the world—four species of this genus are reported: C. brevicauda, C. castanea, C. monohernandezi and C. perspicillata (Mantilla-Meluk, Jiménez-Ortega & Baker, 2009; Zurc & Velazco, 2010).

Taxonomically, the morphological species boundaries of Carollia species have not been clearly determined, some of them being considered as species complexes yet to be resolved (Baker, Solari & Hoffman, 2002; Pacheco, Solari & Velazco, 2004; Solari & Baker, 2006; Jarrín, Flores & Salcedo, 2010). In Colombia, the taxonomy of the genus is of special interest because the identities of some species described for the country are still unresolved (Cuartas, Muñoz & González, 2001; Muñoz, Cuartas & Gonzáles, 2004; Zurc & Velazco, 2010).

Some studies of cranial morphology in Carollia species suggest that size variation is the principal source of morphological plasticity (McLellan, 1984; Jarrín, Flores & Salcedo, 2010), however there is a lack of understanding about the patterns in shape variation. One study reported that skull shape variation was related to environmental fluctuations, and that the relationship was species-specific (Jarrín & Menendez-Guerrero, 2011). Sexual dimorphism is another source of morphological variation that has been discussed, being reported as absent (McLellan, 1984) and present (Jarrín, Flores & Salcedo, 2010).

Up to this point, other important factors (e.g., morphological modularity and integration) that may influence the structuring of morphological variation and generation of morphological diversity have not been investigated in phyllostomids (Jarrín, Flores & Salcedo, 2010; Jarrín & Menendez-Guerrero, 2011).

Morphological integration is the tendency in certain traits within a structure to be correlated in their variation, so that they will co-vary (Klingenberg, 2014). The concept of modularity is related to integration because it describes subsets of traits (modules) that are highly connected (strongly integrated) to one another in comparison to connections between other traits (Klingenberg, 2014). Studies of modularity may clarify how different mechanisms (functional, evolutionary, ontogenetic, environmental or genetic) influence the way in which morphological variation is structured (Klingenberg, 2009; Goswami et al., 2014). A general pattern of cranial modularity based on functional traits is accepted for many mammal species; this pattern distinguishes two different modules: one at the facial region (splanchnocranium), and the other at the posterior region of the skull (neurocranium) (Hallgrimsson et al., 2004; Koyabu et al., 2014). Functional differences between modules are associated with brain developmental processes and muscle insertion in the neurocranium (Reep & Bhatnagar, 2000; Pitnick, Jones & Wilkinson, 2006), and the biomechanics of biting behavior in the splachnocranium (Goswami & Polly, 2010; Wellens & Kuijpers-Jagtman, 2013). In bats, the effect of morphological specializations for echolocation on cranial modularity has been evaluated, concluding that, despite specializations, patterns of modularity remain consistent with those reported for other mammals (Santana & Lofgren, 2013).

Similarly, patterns of mandibular modularity in mammals are described as a response to functional differences between regions in the mandible, reflecting two different modules: the ascending ramus and the alveolar region (Klingenberg & Mebus, 2003; Jojić et al., 2007; Zelditch, Wood & Bonett, 2008; Jojić, Blagojević & Vujošević, 2012). Functionally, the ascending ramus is relevant for muscle insertion and articulation with the skull (Herring et al., 2001), whereas the alveolar region supports the dentition and is associated with food loading and processing (Cox, 2008).

By using this approach it is possible to study cranial and mandibular morphological variation as a unit, evaluating if modularity between both structures is functionally correlated for biting, providing evidence of skull-jaw integration as a functional unit. This correlation for biting is poorly understood, due to the influence that factors like echolocation could have on skull-jaw integration, having been reported only once in mammals (Garcia et al., 2014).

The goal of this study is to provide a quantitative evaluation of cranial and mandibular morphology in Carollia species, specifically focusing on (1) the magnitude and mode of intraspecific shape variation, which is poorly understood, and (2) the influence of the Andes on the distribution of shape variation in populations located in each biogeographic region. Using geometric morphometric methods, we focus explicitly on the quantification of shape variation in Carollia by analyzing trait correlations, typically referred to as the study of modularity and integration. In parallel, by combining geographic and morphologic data we will evaluate the effect of altitudinal barriers (i.e., Andes) on the biogeographic patterns of the morphological variation in this genus.

Evolutionary studies reveal the influence of the Andean orogeny and tropical forest formation in the diversification processes of Carollia (Hoffmann & Baker, 2003; Pavan, Martins & Santos, 2011). Also, the Andes have been identified as a barrier affecting the distribution of morphological variation as a possible consequence of gene flow interruption between populations of the same species (Jarrín & Menendez-Guerrero, 2011). This is especially relevant for C. castanea due to its small body size and lowland distribution. Previous studies proposed that the small size of C. castanea prevented individuals from crossing the Andes and hence altitudinal barriers were hypothesized to restrict gene flow between populations (Jarrín & Menendez-Guerrero, 2011). Studies of the relationship between morphological features, resource partitioning and the coexistence of Carollia species have produced contradictory results, specifically concerning whether limiting similarity determines sympatry or not. York & Papes (2007) found that morphologically distinct species lived sympatrically, whereas more recent study by Jarrín & Menendez-Guerrero (2011) concluded that morphologically similar species cohabited. These inconsistent results raise the question of whether assemblage composition and sympatry in Carollia favors morphologically similar or distinct species (Jarrín & Menendez-Guerrero, 2011).

Materials and Methods

Sample sites and specimen selection

A total of 286 specimens of Carollia (C. brevicauda = 108; C. castanea = 82; C. perspicillata = 96) from 143 different localities in Colombia were evaluated for this study (see Table S1). The criteria for specimen selection were: that only sites with at least one male and one female available were considered, and, to ensure adequate representation of all five biogeographic regions (Caribbean, Pacific, Andean, Amazonian, and Orinoquean) and independence between samples (sites separated by at least 30 km), that one locality only was selected per municipality for each species (Fig. 1).

Figure 1 Map showing geographical distribution of locations sampled for Carollia perspicillata (concentric circles), C. brevicauda (grey circles) and C. castanea (black circles), within the biogeographic regions present in Colombia.

All specimens were obtained from the Instituto Alexander von Humboldt (IAvH-M), Colección Teriológica de la Universidad de Antioquia (CTUA), Instituto de Ciencias Naturales de la Universidad Nacional de Colombia (ICN) and the Museo Javeriano de Historia Natural (MPUJ).

Morphological analysis

Photographs were taken with a Nikon D5100 mounted on a tripod; crania were photographed in ventral view and mandibles in lateral view. In order to optimize and standardize the photographs, focal distance was estimated using the method proposed by Blaker (1976) and different holders were used for crania and mandibles.

Following geometric morphometric principles, landmark configurations were established for crania and mandibles separately using type 1 and 2 landmarks (Bookstein et al., 1985). Modifying the methodology used by Jarrín & Menendez-Guerrero (2011), a total of 15 landmarks were used for the cranium (Fig. 2A), and following previous studies (Zelditch, Wood & Bonett, 2008; Jojić, Blagojević & Vujošević, 2012) 12 landmarks were used for the mandible (Fig. 2B) (see Table S1). Landmark digitalization was performed using TPSDIG version 2.16 (Rohlf, 2010).

Figure 2 Landmark configurations used in this study for the analysis of shape variation of skull (A) and jaw (B).

Generalized Procrustes Analysis (GPA) was performed in order to superimpose landmark coordinates, obtaining the average coordinates of all landmarks in a tangent configuration; this was performed separately for the cranium and mandible datasets (Rohlf, 1990). GPA removes non-shape sources of variation resulting from scaling, rotation and translation (Rohlf, 1999). A tangent configuration is the configuration of landmarks projected from a nonlinear shape space into a tangent space in which parametrical statistical analysis can be performed. Using TPSRELW (Rohlf, 2010), a Relative Warp Analysis (RWA) was performed following the principle of the thin-plate spline technique, which allows the partition of the total variation among all specimens from the tangent configuration in two different components: affine components that describe differences in uniform shape variation (principal warps), and non-affine components that express local variation within the shape (partial warps) (Rohlf, Loy & Corti, 1996).

Relative Warps (RW) are the principal components of a distribution of shapes in a tangent space, comprising the majority of the variation in a few comprehensive components, which are easily visualized using a transformation grid (Rohlf & Bookstein, 2003). RW are non-biological variables used as a representation of affine and non-affine components that describe localized deformations in specific regions of the overall shape, and can be analyzed using conventional statistical methods (Klingenberg, 2013). RW were computed using the partial warps for further statistical analysis.

Patterns of interspecific variation

Interspecific differences in the intraspecific morphological variation were tested with a multivariate analysis of variance (MANOVA) and a paired Hotelling’s test using the RW pooled by species; these analyses were performed using PAST version 2.15 (Hammer, Harper & Ryan, 2001). Squared Mahalanobis distances were used as a measure of morphological distances between species to assess general patterns of variation for all species, P values were corrected with a Bonferroni correction for multiple comparisons α at = 0.05.

In order to detect specific regions where major morphological variation may be focused, RW were visualized using transformation grids for each species, comparing the morphological patterns of variation between each species for the cranium and mandible (Zelditch et al., 2004). Patterns of shape change were depicted using TPSRELW (Rohlf, 2010), and the grids were built with the Principal Components (PC) of the Procrustes coordinates using MORPHO J version 1.04a (Klingenberg, 2011).

Cranial-mandibular integration and modularity

Based on previous findings of functional modularity in mammals (Zelditch, Wood & Bonett, 2008; Monteiro & Nogueira, 2011; Jojić, Blagojević & Vujošević, 2012), two different a priori hypotheses were considered for evaluating morphological modularity, one for the skull and one for the mandible (Fig. 3). The first divided the skull into two functional modules, neurocranium (muscle insertion and brain development) and splachnocranium (feeding and biting behavior); the second divided the mandible also into two functional modules, the ascending ramus (muscle insertion) and the alveolar region (supporting the teeth).

Figure 3 A priori hypotheses tested on C. brevicauda, C. castanea and C. perspicillata for cranial and mandibular modularity.

(A) Cranial modularity divides the cranium into neurocranium (grey-solid lines) and splachnocranium (black-dotted lines). (B) Mandibular modularity divides the jaw into ascending ramus (grey-solid lines) and alveolar region (black-dotted lines).

These hypotheses were evaluated with the Escoufier’s RV coefficient using MORPHO J version 1.04a (Robert & Escoufier, 1976; Klingenberg, 2009; Klingenberg, 2011). This method takes the RV coefficients of the a priori hypothesis and compares it with coefficients of multiple alternate partitions, and hypotheses with coefficient values closer to zero are not rejected. Delaunay triangulations were considered during module construction among landmarks (Berg et al., 2000). For this study we set 10,000 alternate partitions to compare with each a priori hypothesis, and this procedure was applied for each species.

Studying cranial-mandibular integration allowed us to evaluate whether the cranium and mandible together behave as a functional unit, covarying morphologically in their shape (Klingenberg, 2008). To do this, partial least square analysis (PLS) was performed, which explores patterns of covariation between different blocks of variables. RW were pooled by structure (cranium and mandible) and species, performing a PLS for all species where cranium and mandible shape were assigned as different blocks; this analysis was performed using TPSPLS version 1.18 (Rohlf & Corti, 2000).

Geographic patterns vs. morphological variation

RW of each species were pooled, differentiating biogeographic regions (Caribbean, Pacific, Andean, Amazonian, and Orinoquean); this was done for the cranium and the mandible separately. MANOVA and paired Hotelling’s tests were used to assess morphological differences between populations from different geographic regions, and to test if the Andes represent a barrier that divides morphological differences among populations of the same species, separating populations of different biogeographic regions morphologically. P values were corrected with a Bonferroni correction for multiple comparisons α at = 0.05.

Results

Patterns of interspecific variation

The MANOVA differences between species were significant for both cranium (λ = 0.5185; df1 = 18; df2 = 548; F = 11.84; p = 3.05E–26) and mandible (λ = 0.5966; df1 = 18; df2 = 546; F = 8.937; p = 1.68E–1) data sets. All pairwise comparisons were significant with P-values ≤ 1.05E–03.

Based on squared Mahalanobis distances we found C. castanea to be the most morphologically different species, being most distinct in its cranial morphology from the rest of the species (Table 1).

Table 1 Squared Mahalanobis distances between the three species.

Values for the cranium are above the diagonal and values for the mandible below the diagonal.

	C. castanea	C. brevicauda	C. perspicillata	
C. castanea	–	3.7684	2.4021	
C. brevicauda	1.9828	–	1.2940	
C. perspicillata	2.2967	1.2026	–	

As a general pattern, for all species the majority of the variation was concentrated in the neurocranium, around the suture of the occipital and temporal bones, as well as the area comprising the vomer and the palatine (Fig. 4). Each species showed species-specific variation patterns within these regions (Figs. 4A–4C).

Figure 4 Transformation grids for the first Principal Component (PC) of the RWA.

Grids depict intraspecific cranial morphological variation. The grids for all species: C. brevicauda (A), C. castanea (B) and C. perspicillata (C).

For C. brevicauda, the highest deformation in the neurocranium is displaced towards the mastoid due to a constriction of the occipitomastoid suture and the tympanic part of the temporal bone (Fig. 4A). On the other hand, C. castanea exhibited an expansion in the region of the suture towards the occipital and a reduction of the length of the vomer (Fig. 4B). Finally, morphological variation in C. perspicillata was evident in the basicranium, between the foramen magnum and the vomer, and at the occipital and temporal bones. Variation in both regions showed a general contraction of such bones, leading to a general reduction in the length of the neurocranium (Fig. 4C).

Regarding mandibular morphology, the three species varied in the same regions, but the way in which they varied was different between species. Most interspecific variation was concentrated in the middle region of the ascending ramus and the alveolar region (Fig. 5).

Figure 5 Transformation grids for the first Principal Component (PC) of the RWA.

Grids depict intraspecific mandibular morphological variation. The grids for all species: C. brevicauda (A), C. castanea (B) and C. perspicillata (C).

When comparing variation across species, C. brevicauda showed greater variation in the lower border of the ramus, between the condyloid and angular processes (Fig. 5A); for C. castanea the mandibular tooth row and the base of the ramus expanded, resulting in a constriction of the medium region between the ascending ramus and the alveolar region (Fig. 5B). Carollia perspicillata showed the same pattern in the lower border of the ramus, but in this case the mandibular tooth row was shortened, in contrast to C. castanea (Fig. 5C).

Cranial-mandibular integration and modularity

All a priori hypotheses for functional mandibular modularity were not rejected as they had the lowest RV coefficients, dividing the mandible into two different modules (ascending ramus and alveolar region) according to their functional specializations (Table 2). However, 913 different partitions, including the a priori hypothesis, are compatible with the data for C. castanea, which could mean that, although the evaluated hypothesis was not rejected, there are other factors that affect mandibular modularity in this species. Results indicate the a priori hypothesis for cranial modularity was rejected in all cases, finding alternate partitions with lowest RV coefficients (Table 2).

Table 2 RV coefficients for cranial and mandibular modularity analysis for three species of Carollia.

Species	Modularity	hypRV	minRV	alterRV< hypRV	
C. brevicauda	Cranial	0.1724	0.1329	4,013	
Mandibular	0.2583	0.2583	0	
C. castanea	Cranial	0.2793	0.2417	4,822	
Mandibular	0.3366	0.3366	912	
C. perspicillata	Cranial	0.2206	0.2074	1,665	
Mandibular	0.2345	0.2345	0	
Notes.

hypRV are the coefficient values for tested hypothesis, minRV are the minimum RV coefficient values, and alterRV < hypRV is the number of alternate partitions with coefficients lower than those of the tested hypothesis.

Partitions recovered for mandibular modularity had the same structure for all species. Similarly, for cranial modularity, the same general partition pattern, dividing the cranium into two modules representing the neurocranium and the splachnocranium, was recovered. However, the structure of these partitions varied between species, each species having different modularity patterns, and such differences being present in the sphenoidal section of the basicranium (Figs. 6A–6C). Cranial modularity results for C. brevicauda showed that the neurocranium module comprises the zygomatic process of the temporal bone (landmarks 3–10), while the splachnocranium module comprises the palatine (landmarks 9–10) and vomer bones (landmarks 3–9) (Fig. 6A). For C. perspicillata the neurocranium module comprises the zygomatic process of the temporal bone and the vomer and the splachnocranium module comprises the palatine (Fig. 6C). Carollia castanea showed the most distinct modularity patterns where the neurocranium module extends anteriorly covering the zygomatic process of the temporal and the posterior section of the palatine, while the splachnocranium module extends posteriorly covering the vomer (Fig. 6B). For all species the first three dimensions of the PLS explained around 80% (C. brevicauda 78.32%, C. castanea 84.84% and C. perspicillata 76.91%) of the cranial-mandibular morphological integration, R values were always positive (ranging from 0.37 to 0.65), and the coefficient of determination (r2) values corroborated the significance of the results (Table 3).

Figure 6 Patterns recovered for cranial modularity for C. brevicauda (A), C. castanea (B), and C. perspicillata (C), showing the neurocranium (grey-solid lines) and the splachnocranium (black-dotted lines).

Thicker lines and dots highlight the region where modularity varies between species.

Table 3 Values of the first three dimensions of PLS analysis of cranial-mandibular integration for three species of Carollia.

Species	Dimension	R	r 2	Explained variance	Cumulative variance	
C. brevicauda	1	0.4933	7.95E−03	37.74	37.74	
2	0.4125	5.15E−03	24.43	62.18	
3	0.4677	3.40E−03	16.14	78.32	
C. castanea	1	0.5413	2.31E−02	49.34	49.34	
2	0.6555	1.32E−02	28.18	77.52	
3	0.4793	3.42E−03	7.31	84.84	
C. perspicillata	1	0.5131	8.83E−03	42.27	42.27	
2	0.4333	4.61E−03	22.07	64.34	
3	0.3724	2.62E−03	12.57	76.91	

Geographic patterns vs. morphological variation

MANOVA results were not significant for morphological differences in the mandible between specimens of the five biogeographic regions in any of the species; C. brevicauda (λ = 0.6674, df1 = 36, df2 = 354, F = 1.120, P = 0.2971), C. castanea (λ = 0.9243, df1 = 9, df2 = 72, F = 0.6552, P = 0.7461), C. perspicillata (λ = 0.6555, df1 = 36, df2 = 309, F = 1.024, P = 0.4358). For the skull, MANOVA found significant differences between biogeographic regions in C. brevicauda (λ = 0.5182, df1 = 36, df2 = 357.7, F = 1.906, P = 0.0018) and C. perspicillata (λ = 0.3862, df1 = 36, df2 = 309, F = 0.5179, P = 0.0469), and no significant difference was found for C. castanea (λ = 0.5468, df1 = 36, df2 = 260.3, F = 1.264, P = 0.1538).

Paired Hotelling’s test supports these results, finding significance only for C. brevicauda and C. perspicillata. Results revealed that for both species only specimens from the Andean region were different from the rest; Andean specimens of C. brevicauda were statistically different from Amazonian and Caribbean specimens, and for C. perspicillata Andean specimens were different from those of the Pacific region (Table 4).

Table 4 Corrected P values of paired Hotelling’s tests for cranial morphological differences among biogeographic regions.

Above the diagonal are the values for C. perspicillata and below for C. brevicauda.

	Amazonian	Caribbean	Orinoquean	Pacific	Andean	
Amazonian	–	0.2868	0.2586	0.8123	0.1272	
Caribbean	0.0598	–	0.2962	0.2107	0.4694	
Orinoquean	0.5909	0.694	–	0.2715	0.4703	
Pacific	0.0990	0.2038	0.6271	–	0.0121a	
Andean	0.0220a	0.0230a	0.2221	0.3295	–	
Notes.

a Significant values after Bonferroni correction.

Discussion

Patterns of interspecific variation

Results confirmed that despite the presence of intraspecific variation in all species, the mode of this variation differs between species (Jarrín, Flores & Salcedo, 2010). Among these, C. brevicauda and C. perspicillata (larger species) are most similar, and C. castanea (smaller species) is the most divergent (Table 1). This is consistent with phylogenetic analysis in this genus that shows that C. brevicauda and C. perspicillata are sister species and the most recently diversified, while C. castanea is the oldest species (Hoffmann & Baker, 2003).

Previous studies have shown that major cranial morphological variation in these species is present in the neurocranium, specifically in the region that comprises the occipital bone and the squama portion of the temporal bone (Jarrín & Menendez-Guerrero, 2011), supporting our findings of major cranial morphological variation in the occipital and temporal bones (Fig. 4). Quantifying differences in dietary specialization and breadth between species (Dumont, 1999), as well as the specific characteristics of consumed items, such as object hardness and size, could shed some light on the mechanisms shaping the differences found in the patterns of intraspecific variation (Dumont & Piccirillo, 2005).

In phyllostomid bats, mandibular shape has evolved independently of mandibular size, the direction of shape variation being instead associated with diet and feeding behavior (Monteiro & Nogueira, 2011). Frugivorous bats have similar patterns in loading behavior and pressure point resistance in bones related to the masticatory apparatus that differentiate them between hard-heavy-item consuming species (short and flatten rostrum) and soft-light-item consuming species (elongated and narrow rostrum) (Santana, Grosse & Dumont, 2012).

Ecomorphological studies have demonstrated that morphological variation in bats is majorly associated with trophic specialization, and owing to the fact that bat skulls are under selective pressure to reduce their mass (i.e., reduction of skull mass to meet energetic demands of flight), their morphology might be optimized to meet functional demands (Dumont, 2007). Based on this, our findings might reflect interspecific ecomorphological differences in response to biological specializations for optimizing resource exploitation of soft and light items like Piperaceae fruits, one of the principal components of the diet in Carollia (Nogueira, Peracchi & Monteiro, 2009; York & Billings, 2009).

Evidence of niche differentiation and diet specialization for avoiding ecological competition and niche overlap has been reported in phyllostomid bats (Aguirre et al., 2002; Giannini & Kalko, 2004). Species-specific patterns of intraspecific morphological variation found in our study support the hypothesis of interspecific ecomorphological differentiation, which in Carollia is especially evident in sympatric species, where differences in diet breadth and composition have been studied (Lopez & Vaughan, 2007; York & Billings, 2009).

However, given that recent evidence suggests that more historical processes such as niche conservatism also influence the composition of assemblages in phyllostomid bats (Villalobos, Rangel & Diniz-Filho, 2013), to reach a greater understanding of the mechanisms underlying assemblage composition in this genus, it is advised to combine morphometric and phylogenetic approaches (i.e., community phylogenetics). The latter would give a more comprehensive understanding of the role that both historical, and ecological processes have in shaping the structure of modern geographic patterns of coexistence (Villalobos, Rangel & Diniz-Filho, 2013).

Cranial-mandibular integration and modularity

Cranial-mandibular integration was tested to determine whether the structures work together as a functional unit. Hypotheses tested in this study have been successfully studied in other mammals, revealing the importance of functionality in ecomorphological specialization and differentiation in mammals (Klingenberg & Mebus, 2003; Jojić et al., 2007; Zelditch, Wood & Bonett, 2008; Jojić, Blagojević & Vujošević, 2012). Our results indicate that cranial and mandibular modularity has different, independent patterns. Mandibular modularity was the same for all species, so that patterns in this trait were evident at the genus level, while cranial modularity patterns were species-specific. The lack of variation in mandibular modularity is consistent with findings that modularity patterns in the mandible are genetically patterned, which has been suggested to explain the highly conserved module identity (Klingenberg, Leamy & Cheverud, 2004). The variability found for the cranial patterns may align with evidence that cranial modularity can shift on relatively short time scales in relation to selective pressure (Beldade, Koops & Brakefield, 2002; Monteiro & Nogueira, 2010) and requires further, future investigation in the context of Carollia.

Mandibular modularity has so far not been tested in bats and in this first approach our results agree with those reported previously in other mammals, specifically identifying mandibular modularity as a two-module partition defined by functional traits (Klingenberg & Mebus, 2003; Monteiro & Bonato, 2005; Zelditch, Wood & Bonett, 2008; Jojić, Blagojević & Vujošević, 2012). Presence of these modules (ascending ramus and alveolar region) represents differences in functional specializations between different regions of the jaw for biting and food manipulation (Hiiemae, 2000; Badyaev & Foresman, 2004). The shape of the ascending ramus has evolved to support muscle insertion of masseter, pterygoid and temporal muscles which are related to jaw movement and mastication (Herring et al., 2001). The alveolar region specializes in supporting the dentition and loading capacity, which are important for the masticatory apparatus to resist tension-compression forces applied to the bone structure (Cox, 2008). Finding the same results for all species could indicate that ecomorphological plasticity of the jaw does not affect its modularity, also suggesting that this partition is evolutionarily stable and functionally appropriate for the ecology of these species (Koyabu et al., 2011).

Regarding cranial modularity, differences found in module partitions among species could reflect ecological differences in foraging behavior and niche partitioning and their relation with morphological specializations reported for these three species (Giannini & Kalko, 2004; York & Billings, 2009). These modules (neurocranium and splanchnocranium) represent functional specializations in different areas of the skull; the neurocranium exemplifies morphological specializations for muscle insertion and brain development, and the splachnocranium for biting biomechanics and masticatory activity (Hallgrimsson et al., 2004; Goswami & Polly, 2010; Wellens & Kuijpers-Jagtman, 2013). The latter is reported to be in turn related to morphological diversification in the dentition (Santana, Strait & Dumont, 2011) and rostrum (Nogueira, Peracchi & Monteiro, 2009; Santana, Dumont & Davis, 2010; Santana & Dumont, 2011). Other tested hypotheses that evaluated alternative sources of variation that could explain the presence of these modules in bats (e.g., developmental, genetic or ecological) have been rejected, suggesting a strong correlation between evolutionary conservatism in these modules and its functionality (Goswami, 2007; Santana & Lofgren, 2013). Modifications in the neurocranium are associated with differences between trophic guilds in such a way that cranial structure influences functional importance and recruitment of masseter, pterygoid and temporal muscles during biting (Herring et al., 2001). Additionally, the neurocranium is related to brain development that, in bats, co-varies with foraging behavior and mating systems (Pedersen, 2000; Reep & Bhatnagar, 2000; Pitnick, Jones & Wilkinson, 2006).

Functionality of the masticatory apparatus will depend on the correlation between cranium and mandible structure (Hiiemae, 2000), this correlation was evident from the PLS results, which showed that cranial-mandibular integration explained approximately 80% of the shape variation in all species (Table 3). This integration is due to multiple factors that divide the morphological correlation into regions specialized for muscle insertion (neurocranium and ascending ramus) and regions specialized for biting biomechanics (splachnocranium and alveolar region); these regions together comprise the functional and morphological aspects of trophic diversification and fitness in mammals (Freeman, 1998; Cornette et al., 2013). Morphological integration between the neurocranium and the ascending ramus relates to muscle recruitment, and, depending on the feeding behavior and characteristics of the diet, the functional importance of specific muscles will change, altering the morphology of the skull and jaw in order to work as a functional unit and produce the optimal bite force for each species (Santana, Dumont & Davis, 2010). Consequently, it can be deduced that the morphology of the neurocranium and the ascending ramus will vary jointly, forming a component of a functional unit that will correlate with variation in the rostrum, and that is more important in loading capacity and pressure resistance during biting (Cornette et al., 2013).

Rostrum shape variation in rhinolopid bats has been attributed to evolutionary processes of ecological specialization resulting in niche partitioning among ecomorphologically similar species (Santana, Grosse & Dumont, 2012). These processes respond mainly to functional requirements based on an organism’s alimentary and nutritional needs, which relate to shape diversity for exploiting particular resources (Nogueira, Peracchi & Monteiro, 2009; Labonne et al., 2014). The splachnocranium and alveolar region form the rostrum. These modules correlate functionally with biting biomechanics (Dumont & Herrel, 2003), generating functional convergences in load capacity of pressure points in both the cranial and mandibular structures (Herring et al., 2001; Badyaev & Foresman, 2004).

It is established that in these points of pressure the relationship between the proportional importance of tension-compression forces is the same in the cranium and mandible, integrating the two structures (Herring et al., 2001). Accordingly, cranial-mandibular morphological integration found in this study reveals the presence of a functional unit of the skull and jaw, subdivided into two different modules reflecting the functional requirements for both muscle insertion and biting biomechanics (Santana, Dumont & Davis, 2010; Cornette et al., 2013).

Geographic patterns vs. morphological variation

Our findings may be explained on the basis of two hypotheses that reflect different aspects of the evolutionary history of the genus Carollia. Our results reveal: (1) morphological differences at a phylogroup level for these species, which could be an indicator of ongoing processes of speciation, and (2) geographic patterns of morphological variation in these species are influenced by geographic isolation of populations occurring in the Andes.

For this genus, a phylogroup is defined as a group of individuals that share evolutionary history and a geographic location (Hoffmann & Baker, 2003). In C. brevicauda two different phylogroups have been identified. Both are distributed in Colombia: one covers the Andean, Pacific regions and a portion of the Amazonian region; and the second covers the Caribbean region and a portion of the Orinoquean region. Carollia perspicillata includes three different phylogroups, two of which are present in Colombia, one covering the Pacific and Caribbean regions, whereas the other covers the Andean, Amazonian and Orinoquean regions (Hoffmann & Baker, 2003). In Colombia, only one of the four phylogroups described for C. castanea is present; that phylogroup is present in the Pacific region and it is suggested that another phylogroup could be present in a small portion of the Amazonian region (Pine, 1972; Hoffmann & Baker, 2003).

The patterns found in this study fit with the distribution of these phylogroups, rising the hypothesis that morphological differences between phylogroups can be detected based on the geographical distribution of their morphological variation, further suggesting that our results might shed light on ongoing processes of speciation within C. brevicauda and C. perspicillata (Marchiori & Bartholomei-Santos, 2014). This is supported by the idea that for phyllostomid bats the processes of speciation and diversification in the neotropics are related to the orogeny of the Andes (Hoffmann & Baker, 2003; Velazco & Patterson, 2013). However, our results only give preliminary evidence to this conclusion due to the uncertainty of the exact genetic compatibility between our specimens and proposed phylogroups, so it is suggested for further studies to combine both morphometric and molecular techniques to evaluate this particular hypothesis.

It has been postulated that several species in this genus are different species complexes that remain unsolved (Jarrín, Flores & Salcedo, 2010), so our results could provide insight into this topic. Nevertheless, it will be necessary to perform more detailed studies testing the link between intraspecific morphological differences and the distribution of the phylogroups in the neotropics, in order to detect the presence of undescribed species.

Our second hypothesis focuses on intraspecific ecological differences. Limiting similarity has been described as the main factor that determines the composition of species in the genus Carollia; this contends that species that are more similar ecomorphologically will tend not to coexist thereby avoiding competitive exclusion, and hence more morphologically dissimilar species will coexist (York & Papes, 2007). More recent studies have invalidated this hypothesis, showing that morphologically similar species share environmental space, and that dissimilar species coexist less often (Jarrín & Menendez-Guerrero, 2011). Our results agree with those reported by Jarrín & Menendez-Guerrero (2011) in Ecuador, revealing that C. castanea is the species with the most differentiated ecomorphology and distribution of its morphological variation; conflicting with the limiting similarity hypothesis for this genus in the northern Andes. Jarrín & Menendez-Guerrero (2011) propose that the Andes represent a geographic barrier for C. castanea, isolating populations and generating morphological differences between them. As a conclusion, they stipulate that large body size is a buffer that allows large-bodied species to cross the Andes, maintaining the gene flow and morphological similarities among populations. Our study does not support this. Our results are contrary to those from Ecuador in two ways: (1) we found that for larger species (C. brevicauda and C. perspicillata) not only Andean populations are the only ones morphologically differentiated from other populations across the country, but also populations on opposite versants of the Andes are similar; (2) all C. castanea populations across the country showed the same patterns, such that the Andes do not represent a geographic barrier isolating populations from different regions. Based on our results, we hypothesize that only populations present in the Andes are different in their cranial shape from populations in the rest of Colombia. In this way, the northernmost region of the Andes acted more like an independent and isolated environmental region rather than a barrier splitting lowland areas. Inconsistencies between our results and those reported for bats in Ecuador (Jarrín & Menendez-Guerrero, 2011) may be due to environmental differences between the central and northern Andes. The Andes of Ecuador form one single mountain range, while in Colombia the Andes form three mountain ranges, leading to major ecosystem heterogeneity in the interandean valleys of Colombia (Josse et al., 2009). This could represent a wider range of environments to which species may adapt, occupying greater niche diversity without competition (Bloch, Stevens & Willig, 2011; Pereira & Palmeirim, 2013).

Finally, by comparing results from Ecuador with ours we do not support the hypothesis that large body size favors larger species to cross altitudinal barriers, stabilizing genetic pools and morphologies among populations. Our results elucidate that Andean populations of large-bodied species are morphologically different from populations at lower altitudes, which could be a consequence of gene flow interruption between them. Recently, an inverse relation between body size and altitude was discovered in C. perspicillata, where body size decreases along an altitudinal gradient (De Barros & da Fortes, 2014), supporting our conclusion that large species in this genus do not have a competitive advantage in this regard.

Conclusion

Intraspecific shape variation shows species-specific patterns with C. castanea being the most divergent species morphologically, which could indicate ecological differences between species as a consequence of niche partitioning. Strong correlation between the shape of the skull and jaw indicates significant cranial-mandibular morphological integration for all species; this integration corresponds to functional convergences between both structures. Partitions for cranial modularity were species-specific, whereas those for mandibular modularity were the same across all species. Patterns found for cranial modularity indicate that other non-functional factors should be considered when analyzing this feature. In larger species (C. brevicauda and C. perspicillata), Andean populations were cranially morphologically different from other populations, refuting the suggestion that the northern Andes represent a geographic barrier, and instead supporting the idea that the northern Andes represent an independent region that isolates populations occurring there. Finally, and contrary to the idea of large body size acting as a buffer for species in this genus, the smaller C. castanea was the only species that did not show a morphological response to the altitudinal barrier of the Andes.

Supplemental Information

Table S1 Collection information of specimens of Carollia brevicauda, Carollia castanea and Carollia perspicillata used for this study.

Click here for additional data file.

Table S2 Description of landmarks used to evaluate cranium and mandible shape variation.

Click here for additional data file.

Supplemental Information 3 Dataset

Weight matrix of the relative warps for the cranial-mandibular shape variation of C. brevicauda, C. castanea and C. perspicillata.

Click here for additional data file.

We would like to thank Sergio Solari (CTUA), Claudia Medina (IAvH-M), Hugo López and Hugo Mantilla-Meluk (ICN) for granting us access to all the collections and allowing us to review the material. We thank Julio Mario Hoyos (Pontificia Universidad Javeriana) for comments and orientation during the initial stages of the study. Also, we thank Richard Stevens from the TTU for comments and revision of the English language. The first author thanks Laura Castañeda-Gómez for helpful input on earlier drafts of the English language and figures amendment and Suzanne Hand of the University of New South Wales for deep content revision of the final version of the manuscript. Finally, we would like to thank Virginia Abdala, Liliana Davalos and one anonymous reviewer for their helpful comments and questions on earlier drafts of the manuscript.

Additional Information and Declarations

Competing Interests

Author Contributions

Data Availability

Laura A. B. Wilson is an Academic Editor for PeerJ.

Camilo López-Aguirre conceived and designed the experiments, performed the experiments, analyzed the data, contributed reagents/materials/analysis tools, wrote the paper, prepared figures and/or tables, reviewed drafts of the paper.

Jairo Pérez-Torres analyzed the data, contributed reagents/materials/analysis tools, wrote the paper, reviewed drafts of the paper.

Laura A. B. Wilson analyzed the data, wrote the paper, prepared figures and/or tables, reviewed drafts of the paper.

The following information was supplied regarding the deposition of related data:

The raw data all appears in the Supplemental Information of this article.

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
