# Peer review of "Cranial and mandibular shape variation in the genus Carollia (Mammalia: Chiroptera) from Colombia: biogeographic patterns and morphological modularity"

_PeerJ, doi:10.7717/peerj.1197_

## Round 0.1 · original submission · Major Revisions

· Academic Editor

Major Revisions

I have now received two reviews of your manuscript, they find the study very valuable. However, there are significant issues that need to be addressed before it can be published. Please, pay attention to the tittle, maybe it should be more comprehensive. The Introduction will be strengthened if the presentation of the two main themes of the ms – morphological modules and biogeographic processes- is improved to show the reasons to address them together in this ms. I would like you to reorganize the introduction of the ms to clearly set out the order of the themes to be addressed. Additional recommendations of both reviewers are mostly constructive, and attention to their suggestions will serve to improve the discussion of your results.

Reviewer 1 ·

Basic reporting

No Comments

Experimental design

No Comments

Validity of the findings

No Comments

Additional comments

I have read the manuscript by López-Aguirre et al. I think this research is impressive, in all aspects such as sampling, scope, background, design, analyses and interpretations. I have only minor comments and questions that I included directly on the text. I do think maybe the title can be a bit misleading, since the effect of the Andes is just an aspect the authors studied (e.g., the relationship between the Andes and changes in modularity are not discussed). Finally, some citations need to be revised (e.g., Goswami et al. 2014; Pitnick et al. 2006)

Annotated reviews are not available for download in order to protect the identity of reviewers who chose to remain anonymous.

·

Basic reporting

Review of “The effect of altitudinal barriers on morphological variation in the genus Carollia (Mammalia: Chiroptera) from Colombia: patterns of cranial and mandibular modularity and shape variation”. This manuscript examines in detail the cranial variation in 3 species of widely distributed and ecologically important bats, for these reasons it will be of interest to ecologists, functional morphologists and biogeographers. At the moment, this manuscript actually comprises 2 relatively large findings that make the ms difficult to follow: the first has to do with levels of integration and identity of morphological modules, and the second with ecological or biogeographic processes that may explain intraspecific variation in cranial variation. These two findings sit together rather uneasily, ad the first part has to delve into speculation of the functional aspects of mastication and dietary composition that give rise to the modules, while the second takes statistical findings that could correspond to anything (genes, length of tibia, whatever) and maps them out to probable drivers such as Andean orogeny. The main changes necessary to strengthen the paper follow:
1) It would seem that the authors could very well separate this into 2 papers and suffer no loss in the process. However, this could be contrary to some non-obvious ulterior goal. If so, then the structure of the paper could shift so that the introduction followed some kind of logical order. The current intro starts out with taxonomy (which is never revisited in the paper), then the biogeography/competition hypotheses (which correspond to the second half of findings), then modules and masticatory features (which corresponds to the first half of findings. That is, the introduction does not follow the order in which results are discussed, resulting in confusion.
a. On this same topic: the relationship between coexistence and morphology is a fraught one, with many discussions arising in the last decade as the field of community phylogenetics has grown spectacularly. My point is that while cranial morphology can illuminate aspects of the mechanisms of coexistence, it surely misses out other relevant variables.
2) Throughout the modularity section the authors write about “accepted” hypotheses. Surely they mean that specific hypotheses have not been rejected? L228-230 in a null hypothesis testing framework hypotheses are never “accepted”. They are “not rejected”, which has very different implications. L230-233 the meaning of this result is subtly different from what is stated here. The result means 913 different partitions, including the a priori hypothesis are compatible with the data. L235-236 No, hypotheses are never “accepted”. The proposed hypotheses were not rejected hence compatible with the observations. Same issue rest of paragraph. Figure 5: in the null hypothesis testing framework hypotheses are never “accepted”. They are “not rejected”, which has very different implications. I cannot see the difference between gray and black lines here. Maybe this isn’t even the right figure? Use dotted lines or some such. Figure 6: in the null hypothesis testing framework hypotheses are never “accepted”. They are “not rejected”, which has very different implications.
3) L116-125 Cite the map. Map needs serious wok, looks low resolution. The forest background makes it difficult to read, change to clear background, no relief, perhaps just 1,000 m (or some other cut-off elevation) isocline.
4) L377-387 This needs to be discussed with greater caution: there is no matching between the cranial specimens examined and phylogroups in a genetic sense, so the authors are assuming that one kind of variation corresponds to the other, but cannot establish this.
5) L427-432 But in Colombia the Andes disappear at the northern end, generating breaks in the barrier. That is, the patterns found here may in fact be consistent with patterns in Ecuador, it’s just that the Andes are not as much of a barrier in this region compared to Ecuador.
Editorial:
L34 missing “the”
L39 phyllosmotid->phyllostomid
L39-40 what is the evidence for this assertion?
L75 missing period
Table 1: this could be presented in the main text as “all pairwise comparisons were significant with P-values ≤ 1.05E-03“, no need for such a table.
Figure 2: remove the background.

Experimental design

No comments

Validity of the findings

No comments

---

## Round 0.2 · Minor Revisions

· Academic Editor

Minor Revisions

Thank you for your detailed consideration of comments on the previous version of the manuscript. As a result, the paper is greatly improved. I have some minor suggestions (in yellow in the attached ms) and one more question: I do not quite understand the relevance of the political limits in a biogeographic paper. Thus, how is relevant for your study a selection of one locality for municipality for species? Muncipality is an administrative division. Please explain.

---

## Round 0.3 · accepted · Accept

· Academic Editor

Accept

Thank you for editing your manuscript according to the suggestions of the reviewers and myself.